# The Influence of Poultry Manure-Derived Biochar and Compost on Soil Properties and Plant Biomass Growth

**DOI:** 10.3390/ma16186314

**Published:** 2023-09-20

**Authors:** Danuta Dróżdż, Krystyna Malińska, Katarzyna Wystalska, Erik Meers, Ana Robles-Aguilar

**Affiliations:** 1Department of Environmental Engineering, Czestochowa University of Technology, Brzeźnicka 60A, 42-200 Częstochowa, Poland; krystyna.malinska@pcz.pl (K.M.); katarzyna.wystalska@pcz.pl (K.W.); 2Department of Green Chemistry and Technology, Ghent University, Coupure Links 653, 9000 Ghent, Belgium; erik.meers@ugent.be; 3BETA Technological Center Futurlab, Can Baumann Ctra de Roda 70, 08500 Vic, Spain; ana.roblesaguilar@ugent.be

**Keywords:** compost from poultry manure, poultry manure-derived biochar, plant biomass growth

## Abstract

Promising methods for managing poultry manure (PM) include converting poultry manure through pyrolysis to biochar, which can be used for soil applications. The overall goal of this study was to determine the effects of poultry manure-derived biochar and compost on the soil and growth of cherry tomatoes. The biochar obtained at 475 °C was characterized by a relatively high organic matter content of 39.47% and nitrogen content of 3.73%, while it had the lowest C/N ratio of 8.18. According to the recommendations of the EBC, the biochar obtained at 475 °C demonstrated the most beneficial effects in terms of fertilizing potential. The composting of poultry manure with the straw was successful, and the limit of 60 °C was exceeded, which allowed for the hygienization of the compost. The produced compost and biochar are sanitary safe and do not exceed the limits of heavy metal content. The lowest plant biomass was obtained from growing medium A with 3.6 g wet weight (0.24 g dry weight). The measurements of the height of cherry tomatoes showed that growing media D, E, and F allowed the plants to obtain from 602 to 654 mm in height.

## 1. Introduction

Poland has become a leader in the EU in poultry production in recent years. In 2020, Poland produced more than 180 million poultry [1]. Excessive production of poultry also generates significant amounts of waste, mainly litter and poultry manure. The main risks associated with the lack of proper management of poultry manure are emissions of gases and odors, mainly NH_3_, CH_4_, N_2_O, H_2_S, and NO_x_, as well as volatile organic compounds [2]. Excessive ammonia emission is harmful to the health of people and the birds on the farms. It is estimated that annually about 220–280 kg of ammonia is released into the atmosphere from poultry manure produced by 1000 birds [3].

In Poland, poultry manure is used mainly in an unprocessed form and is directly spread on agricultural fields as a rich nitrogen source. Processing methods allowing poultry manure conversion to value-added products with high fertilizing potential can include composting and pyrolysis. These methods will enable the conversion of raw poultry manure into stable materials that can be easily stored, transported, mixed with soil, and distributed in the agricultural fields [4]. With the introduction of new legislation on fertilizing products (i.e., Fertilizing Product Directive from 16 July 2022), it is expected that the interest in such resources as poultry manure to obtain, e.g., organic soil enhancers, will increase. Organic soil enhancers are organic materials that are used to improve soil properties. In general, organic soil enhancers are derived from agricultural residues (i.e., agro-residues) subjected to various biological, chemical, and physical processes [4,5]. 

Poultry manure-based soil enhancers after fulfilling the conformity assessment can be introduced and be available on the EU market [5]. This opens more possibilities for countries with high poultry production and allows significant quantities of poultry manure to be managed.

Therefore, this harmonized legislation creates more opportunities for processing poultry manure into added-value products, which can be used for soil fertilization as growing media. Products such as biochar and compost based on poultry manure can be an alternative to raw poultry manure [2]. These products, compared to raw poultry manure, have limited gaseous emissions, especially ammonia and carbon dioxide, are microbiologically stable, have no significant amounts of heavy metals, and have a beneficial effect on the physical and chemical properties of different types of soil. Poultry manure compost or biochar can be mixed with soil and used as a soil fertilizer. Growing media influence soil aeration, water capacity, and the supply of nutrients (e.g., C, N, P, Na, Ca, etc.) [2,5,6].

The overall goal of this work was to investigate the potential of poultry manure as a source to produce organic soil enhancers, such as poultry manure-derived biochar and poultry manure-derived compost, and determine their physicochemical properties and effects on soil properties and the growth of cherry tomatoes. The presented research article addresses the problem related to the management of poultry manure and the potential for poultry manure-based fertilizing products that can be used for fertilizing soil depleted of, e.g., organic matter. The scope of the work included (1) an analysis of the current state of the art through a literature review, (2) an analysis of the properties of poultry manure sampled from a cage breeding system, (3) laboratory processing of poultry manure through pyrolyzing and composting, (4) an analysis of the properties of the obtained materials intended for soil fertilizing, (5) an analysis of C, N, and P cycles during composting of poultry manure, and (6) an analysis of the effects of the obtained soil enhancers on the soil properties and plant growth.

## 2. Materials and Methods

### 2.1. Substrates

Fresh poultry manure from a caged poultry farm of 30–40,000 *Rosa*-laying hens was sampled for the experiments. Caged poultry breeding is one of the most popular types of breeding in Poland and amounts to about 80% of all types of breeding. 

#### 2.1.1. Fresh Poultry Manure

The fresh poultry manure used in this study (Figure 1) contained residual amounts of eggshells and feathers. 

The fresh poultry manure was promptly analyzed in the laboratory at the Czestochowa University of Technology for physicochemical properties such as pH, nitrogen, organic carbon, available phosphorus, moisture content, and organic matter. The obtained results are presented in Table 1. Sampled poultry manure was also subjected to microbiological analysis *(Ascaris* sp., *Trichuris* sp., *Toxocara* sp., *Salmonella*, and *Escherichia coli*).

The wet bulk density for fresh poultry manure was about 910 kg/m^3^ and the air-filled porosity was 20%. For poultry manure, typical values of wet bulk density are between 800 and 1050 kg/m^3^, and the air-filled porosity in poultry manure is low because of the wet, compacted structure of this substrate [6]. Also, the research conducted by Choi et al. (2001) reported a wet bulk density of raw poultry manure between 900 and 1042 kg/m^3^ and 15–25% air-filled porosity [7]. The fresh poultry manure was characterized by alkaline pH. Poultry manure was characterized by a high concentration of nitrogen of about 8% of dry weight, phosphorus of 75 mg/kg of dry weight, organic matter was 79%, and water content of about 80%. In the study by Singh et al. (2018), the nitrogen content was 5.52% of dry weight, while the C/N ratio was 3.83 [8]. The research conducted by Williams et al. (2013) indicated that phosphorus content can vary between 8 and 34 mg/kg of dry weight, depending on the type of poultry manure and bulking agents that were used for composting. If the poultry manure is mixed with the litter, the phosphorus content is higher than in poultry manure without litter [9]. In another study conducted by Ashworth et al. (2020), the water content was about 75%. This difference was not significant between laying hens and broilers; it depended more on the season in which the poultry manure was collected and the type of breeding [10].

The content of selected heavy metals in the investigated fresh poultry manure was also analyzed. Pb content was <2.00 mg/kg of dry weight, Cd 0.395 mg/kg of dry weight, Cr 11.5 mg/kg of dry weight, and Ni 13.4 mg/kg of dry weight. In the work of Ravindran et al. (2017), poultry manure from 10 different poultry farms was sampled and tested for the occurrence of heavy metals. Pb and Cd were not detected. Cr and Ni were not detected in 4 samples, but in 6 samples the maximum value for Cr was 38 mg/kg, and Ni was 25.7 mg/kg of dry weight [11]. 

In our work, poultry manure was tested for the presence and number of live eggs of intestinal parasites (*Ascaris* sp., *Trichuris* sp., *Toxocara* sp.) at the JARS laboratory in Mysłowice (Poland). These tests were conducted in accordance with the method referred to as (Ae) PB-102/LM ed. 3, dated 25 July 2016. Microbiological analysis showed no evidence of live eggs of intestinal parasites (*Ascaris* sp., *Trichuris* sp., *Toxocara* sp.). *Salmonella* and *Escherichia coli* were not detected in the fresh poultry manure.

#### 2.1.2. Bulking Agent

For composting the poultry manure, wheat straw was selected and used as a bulking agent (Figure 2). Wheat straw is one of the most common bulking agents used in composting. The straw was harvested from a farm located near Czestochowa, Poland.

The wheat straw was cut into 2 cm pieces before the composting process. The wheat straw also was analyzed for physicochemical properties, which are presented in Table 2.

The wet bulk density of the straw was 120 kg/m^3^, and the air-filled porosity was 54%. According to the literature, the bulk density of wheat straw ranges from 97.52 to 177.23 kg/m^3^, and the air-filled porosity from 46.39 to 84.24%. These parameters mainly depend on the place where the straw was harvested, climatic conditions, soil type and location, and method of storage [12].

The wheat straw had a high C/N ratio of 82. The literature reports many studies where wheat straw was analyzed. For example, the wheat straw used for composting organic residues showed a C/N ratio of 69:1, and the moisture content was 14% [13]. Rajput et al. (2018) analyzed wheat straw in terms of changes in physicochemical properties in relation to the initial temperature. The untreated straw reached pH 5.9 while at higher temperatures, such as 180 °C, the pH dropped to 4.8 [14]. Petric et al. (2009) analyzed poultry manure and wheat straw used in the composting process. The wheat straw contained 88% organic matter, the pH was 7.18, the C/N ratio was 88, and the water content was 10% [15].

#### 2.1.3. Soil

The soil used for the plant-growing experiment was collected from an area located near Ghent, Belgium, and had never been treated with animal manure. The soil (Figure 3) was characterized by a loamy–sandy texture and was prone to high water accumulation and clumping into larger aggregates. It was typical soil for Belgium [16].

The soil was left for 2 weeks in a greenhouse at 20–25 °C. After drying, the soil was sieved and used for the plant growth experiment. The physical and chemical properties of the soil are presented in Table 3.

The content of heavy metals such as Pb, Cd, and Ni was not detected. A study on Belgian soils was also conducted by Dassonville et al. (2008) using 36 soil samples from different cities in Belgium for the analysis. The content of N was in the range of 0.69–3.03%, and C was in the range of 0.4–2.4%. The C/N ratio ranged from 3.7 to 19.3: 1, and P ranged from 0.7 to 0.9%. Due to the high use of animal manure to fertilize soils in Belgium, it is difficult to find soil with lower nitrogen content [17].

### 2.2. Composting Mixtures

The composting mixtures were produced by mixing 10 kg of fresh poultry manure with 2 kg of wheat straw. The ratio of poultry manure to wheat straw was 1:0.06 (on a dry basis) and a wet basis was 1:0.26 (on a wet basis). The total weight of the composting mixture was 12 kg. The experiment was performed in two replications (referred to as composting reactor No. 1 and Composting reactor No. 2 and presented individually). 

### 2.3. Growing Media

The following growing media were prepared and used for the experiment: soil (S), compost from composting reactor No. 1 (C1), compost from composting reactor No. 2 (C2), and poultry manure-derived biochar (B). Table 4 presents the description of the investigated growing media.

Due to its high nitrogen content, fresh poultry manure used as a fertilizer generates high ammonia emissions. High emissions of ammonia can have a negative effect on the proper functioning of microorganisms and plant growth. Plants exposed to prolonged ammonium stress do not reach ion balance and had an unfavorable pH homeostasis. Because of these factors, the plant does not develop properly and has lower biomass and less developed roots [18]. For example, the application of a selected treatment to poultry manure can have an influence on physicochemical properties, e.g., a reduction in ammonia during the composting process [2,6,19].

Chen et al. (2020) applied the following doses of the poultry manure compost: 0%, 2%, 4%, and 8%. In terms of plant biomass yield and plant height, the 2% and 4% doses were effective, while the 8% dose did not provide such significant results [20]. Also, Liu et al. (2009) applied poultry manure compost at doses ranging from 0.6% to 2.5% [21]. Meier et al. (2017) used 0.5% and 10% biochar from poultry manure, and they reported beneficial effects on plant biomass. There was a three-fold increase in plant biomass in the sample with 0.5% biochar compared to the control [22]. Masud et al. (2020) studied the effect of poultry litter and poultry manure biochar at doses of 0.5%, 1%, and 1.5% on maize growth. The growth of plant biomass was more beneficial with the addition of biochar, especially at the doses of 1 and 1.5%. The doses of the investigated soil enhancers selected for the experiment constituted the average value adopted from the performed literature review [23].

### 2.4. Methods

This study was conducted at the laboratory of the Department of Environmental Engineering, the Częstochowa University of Technology (Częstochowa, Poland) and the Department of Green Chemistry and Technology, Ghent University (Ghent, Belgium).

#### 2.4.1. Physicochemical Analysis

The following analyses were performed to determine the physicochemical properties of fresh poultry manure, compost from poultry manure, and biochar from poultry manure. All samples were tested in 3 replications. These substrates were tested for moisture content (MC), which was determined based on the norm PN-75/C-04616.01 [24]. MC was determined by drying the samples at 105 °C to constant weight. Organic matter (OM) was determined based on the norm PN-75/C-04616.01. The content of organic matter (OM) was determined by the loss on ignition of the dried mass in a muffle furnace at 550 °C for 3 to 5 h. The bulk density was measured based on the mass in known volume. Air-filled porosity was measured based on bulk density, moisture content, organic matter, and well-known values of water, ash, and organic matter density from the following calculation obtained from an article written by Malińska and Richard, 2006 [25].

The total nitrogen (TN) was determined by the Kjeldahl method according to the norm PN-Z-15011-3 [26]. The phosphorus (P) content was determined based on the methodology of laboratory analysis of soils and plants according to “Methodology of laboratory analysis of soils and plants” written by Karczewska A. and Kabała C. (2008). The content of phosphorus was determined by the spectrophotometric method (Hach Lange DR 5000) with ammonium molybdate. The total organic carbon (TOC) was determined according to Jarvie et al. (1991). The content of carbon in solid samples was analyzed with an organic carbon analyzer (TOC) (Carbon Analyzer Multi N/C 2100, Analytik jena; high-temperature incineration with detection IR, Jena, Germany) at high temperature (1200 °C) oxidation in the stream synthetic air. The pH was determined based on the norm PN-EN 15933:2013-02E [27] by a laboratory pH meter, Elmetron CPC-505. Electrical conductivity (EC) was determined based on the norm PN-EN 27888:1999 [28] by Elmetron CPC-505. Ammoniacal nitrogen was determined based on the norm PN-76/C-04576/01 [29]. Organic carbon (Corg) was determined based on the norm PN-EN 15936:2013-02E [30]. The concentrations of the following elements, i.e., P, S, Ca, Mg, Na, K, Al, Cr, Cu, Fe, Mn, Pb, Zn, Cd, Co, and Ni, were determined with Inductively Coupled Plasma Optical Emission Spectrometry (ICP-OES). The analyses were performed according to the Regulation of the Minister of Agriculture and Rural Development of 18 June 2008 on the implementation of certain provisions of the Act on fertilizers and fertilization from 2007 [31]. These analyses were performed by an ICP OES Vista-MPX analyzer (Richmond Scientific, Chorley, UK).

#### 2.4.2. Production of Soil Enhancers

The soil enhancers were obtained by converting poultry manure through composting into compost and pyrolysis into biochar. The obtained soil enhancers were analyzed for C_org_, N, P_2_O_5_, MC, SO, pH, and EC content, as well as nutrient (Ca, Mg, Na, K, Al, Fe, Mn, Zn, Co) and heavy metal content (Cr, Cu, Pb, Cd, Ni).

Microbiological tests were also performed in the JARS laboratory in Mysłowice (Poland), where raw poultry manure and compost from poultry manure were tested for the presence and number of live eggs of intestinal parasites (*Ascaris* sp., *Trichuris* sp., *Toxocara* sp.). These tests were conducted in accordance with (Ae) PB-102/LM ed. 3, dated 25 July 2016. The microbial analysis for *Escherichia coli* and *Salmonella* was conducted at the Czestochowa University of Technology according to the norms, Pr PN-Z-19000-2 [32] and PN-Z-19000-1 [33], respectively.

##### Composting

The prepared composting mixtures were composted in the laboratory composting reactors for 40 days and after that, the composting mixtures were left to mature for 5 months. The aeration rate during composting was at the level of 35–40 dm^3^/h. The composting mixtures after two weeks were removed from the composting reactors for mixing and sampling. Czekała et al. (2016) also conducted mixing in half of the composting process for improving the aeration and taking samples for the analysis. The temperature in the composting reactors was measured daily [34]. During composting, the composting mixtures were sampled at the beginning, middle, and end of the composting and after 5 months of maturation. Figure 4 presents the matured, dried, and ground poultry manure-derived compost.

##### Pyrolysis

Poultry manure was thermally converted in the laboratory pyrolysis furnace. Pyrolysis was carried out with an inflow of 5 L/min of nitrogen supplied to the furnace. The selected heating temperatures were the following: 475, 575, 675, and 775 °C. Heating of the poultry manure samples took 120 min, and the retention time was 60 min. The study was performed according to the process parameters described by [35]. Figure 5 presents the final product, i.e., ground poultry manure-derived biochar.

#### 2.4.3. Preparation of the Growing Media and Plant Growing Experiment

Each growing medium (A-F) was prepared in five replicates. About 1000 g of each growing medium was transferred into a pot to run the plant growth experiment.

Cherry tomatoes were selected for the plant-growing experiment. Seeds were first sown into a growth tray that had been previously filled with moist medium (coconut fiber/peat mixture). The growing was carried out in the phytotron chamber at a temperature of 23–26 °C. The seeds started to germinate after 4 days, and after 2 weeks they were prepared for repotting into pots with growing media (they grew to a height of 3.5–4 cm). After repotting, the plants were placed in the phytotron chamber. 

### 2.5. Experimental Setups

Three experimental setups were used to conduct the laboratory experiments, i.e., the laboratory composting setup and biochar production system (located in the laboratory at the Częstochowa University of Technology) and the plant growing system in the phytotron chamber (located in the laboratory at Ghent University). 

#### 2.5.1. Composting Setup

The laboratory composting setup (Figure 6) consisted of the insulated laboratory composting reactors with 60 L of volume each, the forced aeration system with the oxygen flow regulators, the temperature probe, and the containers for collecting leachate and condensate, which also served as the system for capturing gases from the outlet air, i.e., NH_3_ and CO_2_. The leachate and condensate were collected but not used for further experiments due to insufficient quantities [19,34].

#### 2.5.2. Biochar Production System

The laboratory biochar production setup (Figure 7) was able to reach a maximum temperature of up to 1100 °C. The pyrolysis reactor (PRWS100x780/11; manufactured by the Czylok company from Jastrzębie-Zdrój, Poland) had the ability to select the temperature and amount of nitrogen relative to the substrate used.

#### 2.5.3. Laboratory Plant Growing System

The plant growth experiment was conducted in a phytotron chamber. The chamber had a controlled temperature and artificial lighting. The plants were exposed to light for 16 h per day, as recommended in the literature to avoid the stress of insufficient light [36]. The specification of the lamps was as follows: Sylvania gro-lux f36W/gro t8 bulbs of 8500 K, 1200 mm long, and 28 mm wide (Figure 8). Light distribution on the shelves was controlled using a Quantum Model MQ-500 m with a separate sensor.

## 3. Results and Discussion

### 3.1. Properties and Fertilizing Potential of the Obtained Soil Enhancers

Soil enhancers obtained through biological, chemical, and physical processes are required to comply with several requirements of the Regulation of the European Parliament and of The Council (EU) 2019/1009 of 5 June 2019. A soil enhancer is characterized by organic matter content, organic carbon content, water content, heavy metals, and microbiological tests.

#### 3.1.1. Poultry Manure-Derived Compost

The process of poultry manure composting was analyzed in detail to have a better understanding of C, N, and P cycles. This section presents the results of the laboratory composting study, including the characteristics of the composting mixtures and mature compost and microbiological analyses of poultry manure-based compost.

##### Temperature Evolution during Composting

The temperature during the 40-day composting process was measured daily. During the composting process, a maximum value of 61.2 °C was observed in composting reactor No. 1 and 60.1 °C in composting reactor No. 2. The maximum temperature values were observed on the 3rd and 4th day of the composting trial, which is typical for proper composting. For example, Czekała et al. (2016) and Dróżdż et al. (2020) observed the maximum increase in the temperature during the first 5 days from the start of composting [19,34]. Figure 9 presents the temperature evolution during the 40-day composting process.

Based on the temperature evolution, the composting process of poultry manure with wheat straw proceeded in a proper way, which is typical for laboratory composting in closed vessels with forced aeration. A temperature >60 °C obtained during the composting process also reduced the growth of pathogens. The temperature sufficient to reduce *Escherichia coli* and *Salmonella* colonies is 47.5 °C [2,19,36,37,38]. Similar values were also obtained by Petric et al. (2008) and Czekała et al. (2016), who prepared composts from poultry manure and wheat straw. The highest value of the composting mixture was from 64.6 and 69 °C [15,34]. 

##### Microbiological Analysis of the Obtained Compost

The composting mixtures after the 40 days of composting sampled from composting reactors No. 1 and 2 were tested for the presence and number of live eggs of intestinal parasites *Ascaris* sp., *Trichuris* sp., and *Toxocara* sp. The analysis presented the absence of *Ascaris* sp., *Trichuris* sp., and *Toxocara* sp. The composting mixtures were also tested for the presence of *Escherichia coli* and *Salmonella,* which were not present in the composting mixtures. 

The Regulation of the European Parliament and The Council (EU) 2019/1009 of 5 June 2019 allows *Escherichia coli* or *Enterococcaceae* in soil enhancers in the permissible amount of 1000 CFUs (Colony Forming Units) in 1 g or 1 mL of a test solution. On the other hand, it does not allow the occurrence of *Salmonella* spp. This research confirmed that the hygienization of the compost was performed properly and that temperatures above 60 °C were sufficient for a reduction in pathogenic microorganisms [39]. The effect of high temperature (70 °C) reduces *Salmonella* within an hour from fertilizer products [5]. 

##### Heavy Metal Content in the Obtained Compost

The obtained results from the composting mixtures and compost from composting reactors No. 1 and 2 were compared to the regulation of the Minister of Agriculture and Rural Development of 18 June 2008 on the implementation of certain provisions of the Act on fertilizers and fertilization from 2007. The permissible values should not exceed <100 mg/kg dry matter for chromium, <5 mg/kg dry matter for cadmium, <60 mg/kg dry matter for nickel, and <140 mg/kg dry matter for lead. The results obtained from the tests for the content of heavy metals confirmed that the compost did not exceed the permissible limits.

The Cu content in an organo-mineral fertilizer must not exceed 600 mg/kg dry matter, and the Zn content in an organo-mineral fertilizer must not exceed 1500 mg/kg dry matter [5]. The results obtained from the tests for Cu and Zn confirmed that the composting mixtures and compost did not exceed the permissible limits [5].

#### 3.1.2. Poultry Manure-Derived Biochar

Poultry manure-derived biochar was produced by pyrolysis at four temperatures: 475 °C, 575 °C, 675 °C, and 775 °C. The biochar yield from one kilogram of poultry manure at temperatures of 475 °C, 575 °C, 675 °C, and 775 °C, was 52.8%, 43.9%, 42.6%, and 40.2%, respectively. Similar results regarding the decrease in the yield of biochar from poultry manure with increasing pyrolysis temperature were reported by Bavariani et al. (2019) and Sobik-Szołtysek et al. (2021). The yield of biochar production depends mainly on the used type of pyrolysis substrate [35,40].

##### Comparison of Selected Properties of the Obtained Biochar with the EBC Guidelines

The biochar that will be used for soil fertilizing should be safe for the environment. To assess and obtain a quality certification that conforms to safety standards, there are several organizations on the market that certify plant-based biochar [41]. These are the IBI Biochar Standards, European Biochar Certificate (EBC), and Biochar Quality Mandate (BQM). The quality requirements for biochar input into the soil include the content of heavy metals (As, Cd, Cr, Cu, Pb, Hg, Ni, and Zn), polycyclic aromatic hydrocarbons (PAH-16), polychlorinated biphenyls (PCB-7), furans and dioxins (PCDD/F), dry matter content, pH, total organic carbon (TOC), nitrogen (N), potassium (K), total phosphorus (as P_2_O_5_), total calcium (Ca), and magnesium (Mg) [42]. According to the guidelines of REFERTIL Recommended Biochar Quality (2018), a set of physicochemical analyses of poultry manure-derived biochar were performed, and the obtained results were compared with the reference values for The European Biochar Certificate (EBC). The properties of biochar obtained at selected pyrolysis temperatures are presented in Table 5.

Several researchers [35,43,44,45] have confirmed that the higher the process temperature was, the more ash was present in biochar. Consequently, the high content of ash increases the pH of the obtained biochar. Domingues et al. (2017), while studying biochar derived from the mixture of poultry manure and coffee bean shells, observed that biochar at 350–450 °C has a lower specific surface area and higher water absorption. Furthermore, it can absorb N-NH_4_ and reduce nitrogen loss during leaching [46]. This means that biochar from poultry manure could be also used as an additive to composting mixtures with high nitrogen content materials, and the obtained compost could be applied as soil enhancers [20,47]. Chen et al. (2020) used the addition of biochar from poultry manure during composting of poultry manure. They found that the addition of 4 and 6% of biochar reduced the release of greenhouse gases from the composter such as methane by 19–27%, nitrous oxide by 9–55%, and ammonia by 24–56%. The addition of biochar had a positive effect on the extension of the thermophilic phase and maturation of the poultry manure compost [20]. Ronix et al. (2021) studied the effect of poultry manure biochar addition on soil properties and plant growth. The addition of 5% biochar increased organic matter content and improved soil structure. Biochar had a positive effect on cabbage growth [48]. 

In the presented experiment, the pH ranged between 12 and 13. According to the European Biochar Certificate (EBC), (2022), the pH of biochar should range from 6 to 10. However, this parameter applies to biochar from plant biomass and not animal-origin biomass [49]. According to the literature, the pH for biochar obtained from animal manure is in the range of 7–13 [35,50,51,52,53]. In terms of heavy metals, their content in biochar from poultry manure did not exceed the standards set by the ECB, (2022) and the Regulation of the Minister of Agriculture and Rural Development of 18 June 2008 on the implementation of certain provisions of the Act on fertilizers and fertilization from 2007.

Based on the presented properties of the obtained biochar, the biochar obtained at 475 °C was selected as a soil enhancer to be used as an additive in the growing media. This biochar was characterized by the lowest pH, higher organic matter content, and micro- and macro-nutrients. The selection of biochar from the lowest pyrolysis temperature (475 °C) was also determined by the fact that there is limited knowledge about the effects of biochar from poultry manure obtained at the temperature of 475 ℃ on soil properties and plant growth. This problem was also noted by Hossain et al. (2021) [53]. 

### 3.2. Effects of the Obtained Soil Enhancers on Soil Properties

In general, soil enhancers are expected to have a positive effect on soil properties. Soil enhancers reduce bulk density in the soil and contribute to the stability of soil aggregates and soil aeration. Soil enhancers are also expected to facilitate the uptake of elements that are in forms not available to plants, in particular phosphorus and nitrogen [54,55]. The results obtained from the physicochemical analyses of the investigated growing media, i.e., soil enhancers in the form of compost and biochar mixed with soil, are presented in Table 6.

Mixing the soil enhancer with the soil with a pH of 6.99 increased the pH of all the growing media. The most significant increase in pH was observed for growing medium D, i.e., the mixture of soil and biochar from poultry manure where the pH was 8.10. High pH is typical for poultry manure-derived biochar, and the pH can range from 7 to even 13 [20,35,40]. Adding compost 1 and 2 and biochar increased the content of N, C, OM, and micro- and macro-elements in growing media B-F. 

The same conclusions regarding the increase in the nutrient content after the application of soil enhancers based on poultry manure in soil were noted by Dail et al. (2007), Jeffery et al. (2011), Ghaly et al. (2012), Ghaly and MacDonald, (2012), Crane-Droesch et al. (2013), Purnomo et al. (2017); Tańczuk et al. (2019), Ansari et al. (2019), and Jindo et al. (2020) [3,56,57,58,59,60,61,62]. 

Lentz et al. (2012) used poultry manure-derived biochar as a soil enhancer by mixing soil and producing a growing medium with a 1.5-fold increase in Mn content, a 1.40-fold increase in C content, and a 1.2–1.7-fold increase in other nutrients compared to the control (soil only) [63]. Adekiya et al. (2019) applied poultry manure-derived biochar, which also had a positive effect on the soil, especially in organic matter content, which increased by 5–30% [51].

### 3.3. Effects of the Obtained Soil Enhancers on the Growth of Cherry Tomatoes

The effects of the obtained growing media after the addition of the investigated soil enhancers were determined from the analysis of growing cherry tomatoes in the 6-week pot experiment.

#### 3.3.1. Effects of Soil Enhancers on Plant Growth

The experiment was documented with photos presented in Table 7. The growth of cherry tomatoes was monitored for 6 weeks until blooming. 

A demonstrable growth effect was observed for growing media C, D, E, and F. The increase in plant biomass was due to the combination of compost and biochar from poultry manure, which were rich in organic matter, supporting soil microbial activity and plant growth. Due to physical and thermal processes, the compost and biochar from poultry manure had a stable structure, limited gas emissions, and a high content of micro- and macro-elements. All these elements contributed to improving soil properties [55,64]. The beneficial effect of applying poultry manure compost is confirmed by Abdelhamid et al. (2004), who applied 20 to 200 g of poultry manure compost per 1000 g of soil. They observed that 20 g of poultry manure compost per pot was enough to see a significant increase in plant biomass (*Faba bean*) [65]. A similar situation was observed by Revell et al. (2012), who applied 5% of poultry manure-derived biochar to sandy soil in which lettuce and peppers were planted. They observed the growth of both plants to be about 50% higher than in soil without additives. They also confirmed that the poultry manure biochar increased soil pH and phosphorus content. The biochar also had a positive effect on water retention in the sandy soil, which increased from 15% to 27% [66].

The use of growing medium F resulted in the highest plant biomass (Figure 10). On average, the plants reached about 41 g of wet weight (3.38 g of dry weight). Growing medium F was a mixture of soil, poultry manure-derived biochar, and compost from composting reactor No. 2. Revell et al. (2012) and Sikder et al. (2018) observed that the addition of poultry manure biochar has a significant effect on the obtained plant biomass compared to the application of dried poultry litter [66,67]. Similar conclusions were also reached by Bhattarai et al. (2015), who applied poultry manure-derived biochar in pea (*Pisum sativum* L.) growth [68].

The lowest plant biomass was obtained from growing medium A with 3.6 g wet weight (0.24 g dry weight). Ojeniy et al. (2008) and Usman (2015) also noted the positive effect of poultry manure on tomato growth compared to the soil without any treatments (control) [69,70]. Most of the plants consist of almost 90% water, so the mass obtained after drying can be several times lower [71]. 

The measurements of the height of cherry tomatoes showed that growing media D, E, and F, i.e., the mixture of soil, compost, and biochar, allowed the plants to obtain from 602 to 654 mm in height. Similar results were obtained by Musa et al. (2020), where tomatoes obtained a height of 400–450 mm after using dried poultry manure and biochar from poultry manure. Compared to the soil without additives, the height of the tomatoes did not exceed 300 mm [72].

#### 3.3.2. Changes in the Growing Media after the Completion of the Plant Growth Experiment

All the results obtained after the 6-week plant growth experiment are presented in Table 8.

The increase in the content of organic matter (OM), nitrogen (N), and carbon (C), after the experiment was completed, was observed in growing media D, E, and F. This situation was due to the fact that it contained biochar and poultry manure compost. This growing medium allowed the retention of nutrients in the soil and prevented their significant leaching and emission into the atmosphere. A similar situation was reported by Lehmann et al. (2007), where 2.5% biochar was added, allowing the soil to retain 12% more nitrogen than in soil without additives [73]. However, Zhan et al. (2015) observed that the addition of biochar resulted in the retention of 27% more carbon and 75% more nitrogen than in the soil without any additives after the pot experiment was completed. When the carbon content increased, it could be related to roots that remained in the soil after the plants were removed from the pots [74]. This situation can increase the results of carbon content. Also, in the research conducted by Dróżdż et al. (2020), an increase in carbon content in the growing media after the plant growth was observed [19]. 

There was also a significant loss of sodium in all growing media. This is related to the fact that sodium is naturally rinsed and evaporates with water on the surface of the pot. It is not a key component for plant growth, but excessive amounts of Na can cause the inhibition of plant growth [75]. 

### 3.4. Characteristics of the Collected Cherry Tomato Plant Biomass after the Completion of the Plant Growing Experiment

The collected cherry tomato plants were analyzed for their chemical composition. The obtained results are presented in Table 9.

According to the literature, plants should contain 3 to 4% nitrogen in above-ground tissues. Nitrogen in plants comes from fertilizers, growing media, nitrogen in the soil, nitrogen from the atmosphere, and water. Microorganisms convert inorganic forms of nitrogen NH_4_⁺ and NO_3_− into forms that are available by immobilization [76].

It can also be observed that the biomass of plants (aboveground part) from growing media B, C, D, E, and F presented higher N content than the biomass of plants from growing medium A (soil only). However, the biomass of plants from growing media B, C, D, E, and F contained less Ca, Mg, Na, and K than plants from growing medium A. This relation can be due to the fact—according to Feng et al. (2020)—that plants absorbing nitrogen from soil low in this compound accumulate more nitrogen in their roots, while Ca, Mg, and Na ions are accumulated in plant biomass [77].

## 4. Conclusions

Based on the results obtained from this study, the following conclusions were formulated:Poultry manure-derived biochar and compost are microbiologically and environmentally safe in terms of heavy metals and have no significant emissions, especially ammonia and carbon dioxide;Biochar obtained at 475 °C has the most beneficial parameters in terms of fertilizing potential compared to biochar obtained at temperatures of 575–775 °C, according to the recommendations of the EBC;The effect of soil additives, i.e., compost and biochar from poultry manure, had a beneficial effect on the growth of cherry tomatoes;Adding biochar to growing media D, E, and F resulted in an increase in pH (7.55–8.00) compared to growing medium A (6.99) used as the control. Adding the obtained compost to growing media B, C, E, and F resulted in an increase of 43–65% in organic matter, 42–60% in N, 40–60% in C, and 37–66% in P compared to growing medium A.

The results from this research can contribute to the advancement of the state of the art by providing a better understanding of the properties of soil enhancers derived from poultry manure, in particular biochar derived from poultry manure, and their effects on soil properties and plant growth.

Further research on the potential of poultry manure to produce soil enhancers should include an overall assessment of the environmental and economic impact of applying poultry manure-derived soil enhancers to close the C, N, and P cycle, as well as the effects on different types of soils under various climatic conditions and selected plants.

## Figures and Tables

**Figure 1 materials-16-06314-f001:**
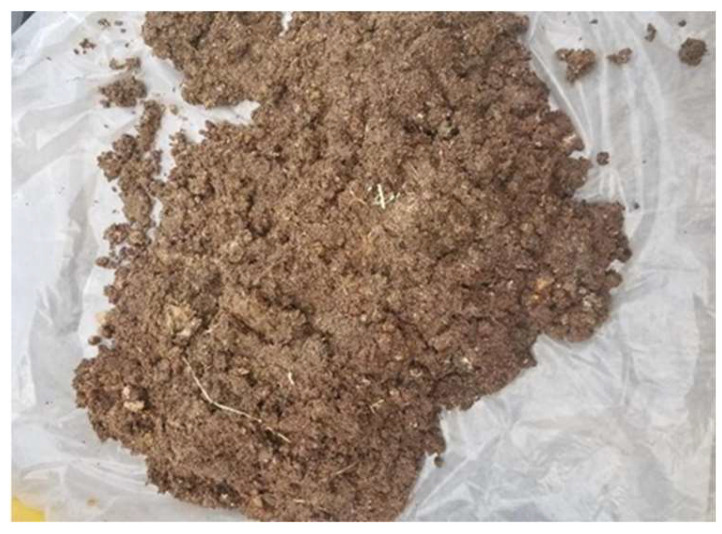
Fresh poultry manure.

**Figure 2 materials-16-06314-f002:**
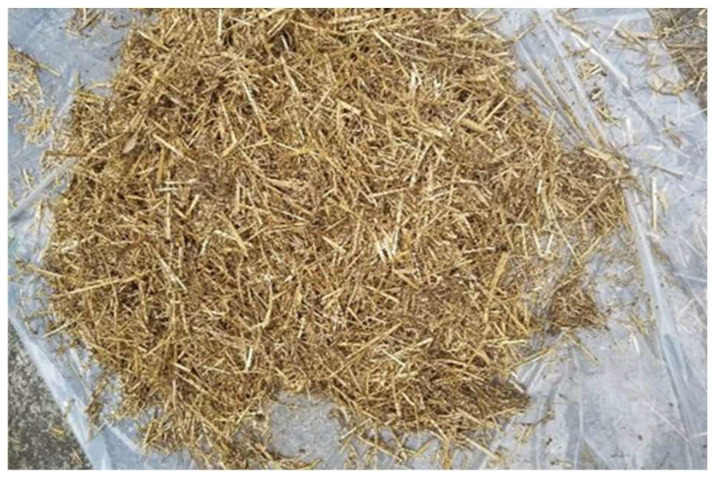
Wheat straw is a bulking agent used for composting poultry manure.

**Figure 3 materials-16-06314-f003:**
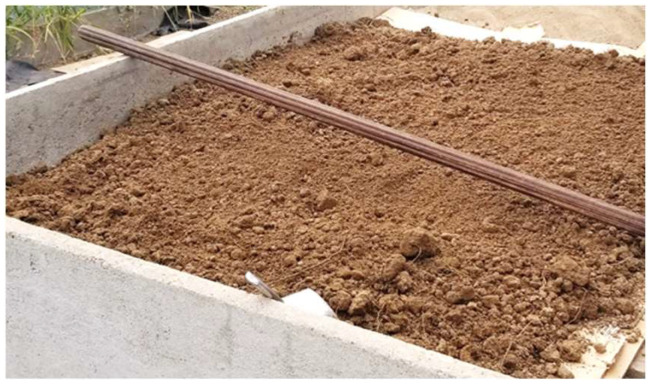
Soil used for the plant growth experiment.

**Figure 4 materials-16-06314-f004:**
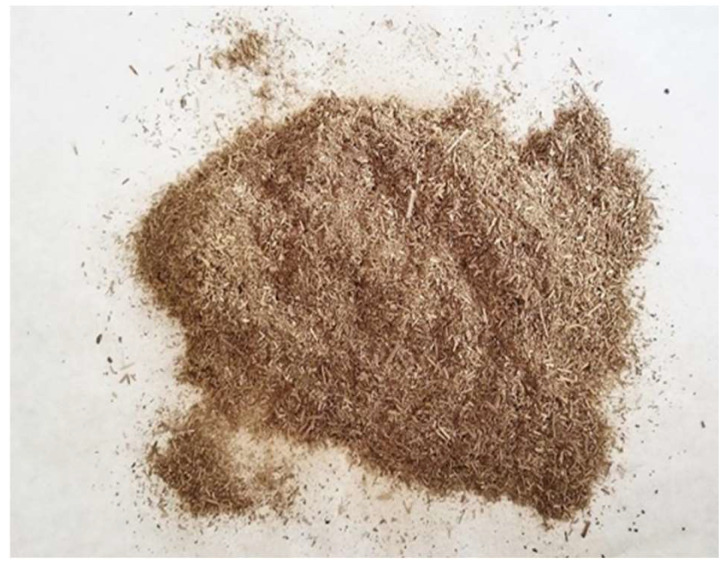
Poultry manure-derived compost.

**Figure 5 materials-16-06314-f005:**
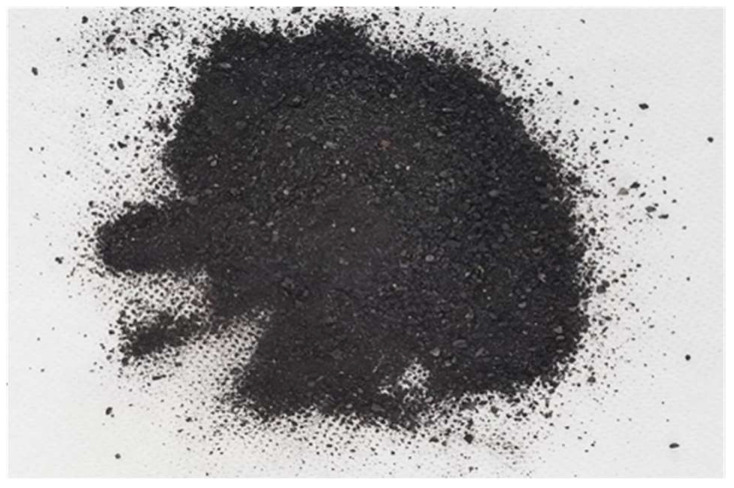
Poultry manure-derived biochar.

**Figure 6 materials-16-06314-f006:**
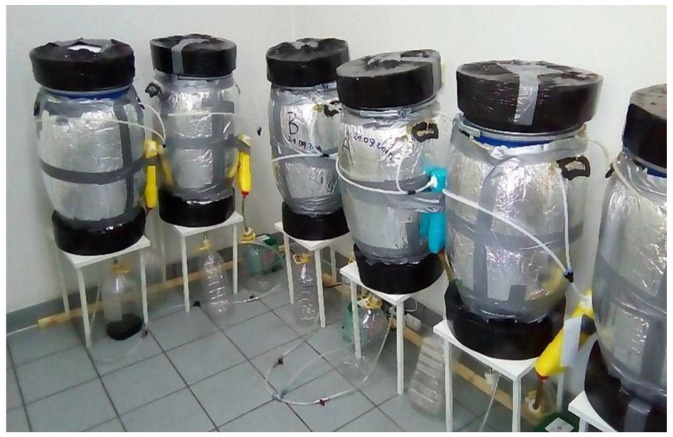
Laboratory composting setup.

**Figure 7 materials-16-06314-f007:**
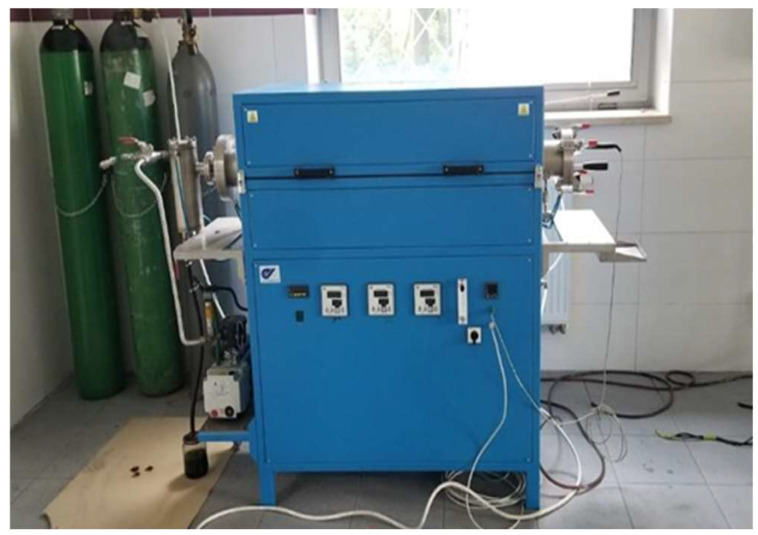
Laboratory pyrolysis furnace for the biochar production system.

**Figure 8 materials-16-06314-f008:**
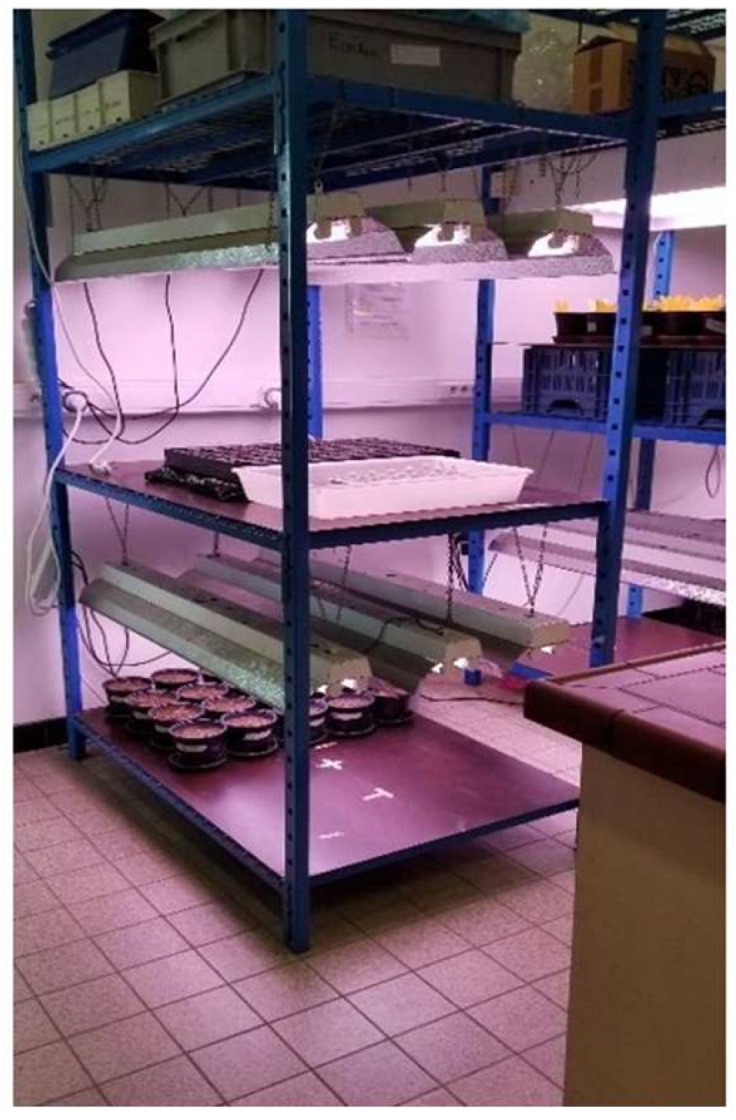
The layout of the cherry tomatoes in the phytotron chamber during the plant growth experiment.

**Figure 9 materials-16-06314-f009:**
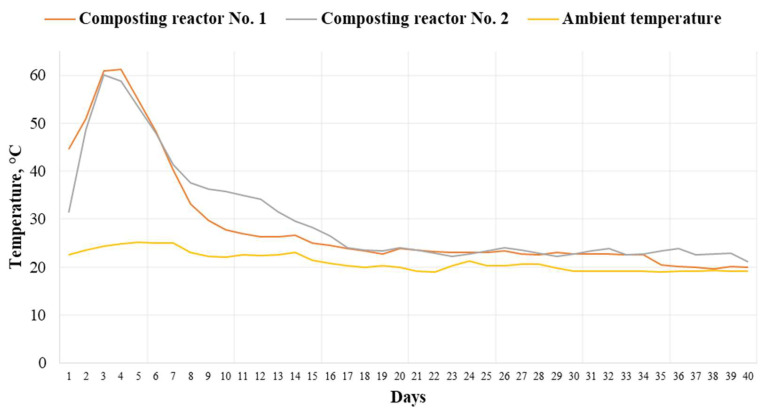
Temperature evolution during the 40-day of composting process in composting reactors No. 1 and 2.

**Figure 10 materials-16-06314-f010:**
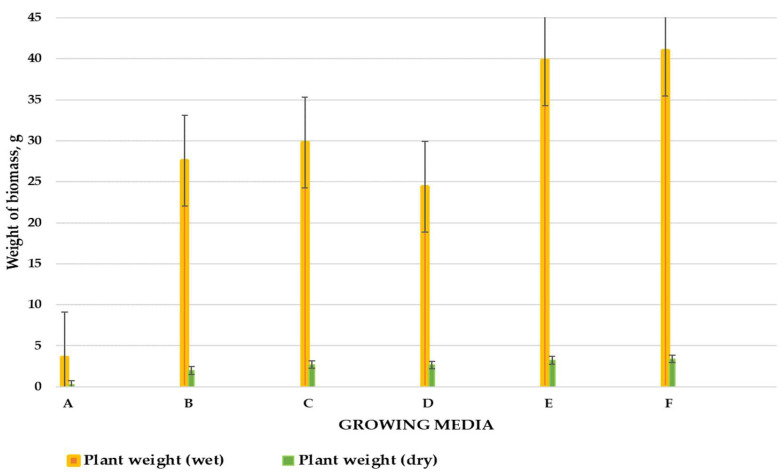
The mass of fresh and dried plants after 6 weeks of growth.

**Table 1 materials-16-06314-t001:** Selected characteristics of the fresh poultry manure.

Parameters	pH	N	C_org_	P_2_O_5_	C/N ^1^	MC ^1^	OM ^1^
Units	-	%	%	mg/kg	-	%	%
Poultry manure	7.51 ± 0.19	7.91 ± 0.01	42.32 ± 0.02	74.70 ± 0.17	5	78.79 ± 0.41	74.43 ± 0.27

^1^ C/N—ratio of carbon/nitrogen; MC—moisture content; OM—organic matter.

**Table 2 materials-16-06314-t002:** Selected properties of the wheat straw.

Parameters	pH	N	C_org_	P_2_O_5_	C/N	MC	OM
Units	-	%	%	mg/kg	-	%	%
Wheat straw	6.97 ± 0.21	0.45 ± 0.02	36.84 ± 0.03	2.50 ± 0.23	82	7.94 ± 0.45	98.47 ± 0.25

**Table 3 materials-16-06314-t003:** Selected properties of the soil used for the plant growth experiment.

Parameters	pH	N	C_org_	C/N	MC	OM
Units	-	%	%	-	%	%
Soil	6.99 ± 0.24	0.45 ± 0.03	0.8 ± 0.13	10	1.54 ± 0.25	2.56 ± 0.64

**Table 4 materials-16-06314-t004:** Description of the growing media.

Name	Type of Growing Media	Ratio (Dry Matter)	Composition of Growing Media
A	S (control)	1	100% soil
B	S + C1	1:0.03	3% compost from composting reactor No. 1 was added to the soil
C	S + C2	1:0.03	3% compost from composting reactor No. 2 was added to the soil
D	S + B	1:0.005	0.5% of biochar was added to the soil
E	S + B + C1	1:0.005:0.03	0.5% biochar and 3% compost from composting reactor No. 1 was added to the soil
F	S + B + C2	1:0.005:0.03	0.5% biochar and 3% compost from composting reactor No. 2 was added to the soil

**Table 5 materials-16-06314-t005:** Comparison of selected properties of the obtained biochar with the EBC guidelines.

Parameters	Units	Biochar from 475 °C	Biochar from 575 °C	Biochar from 675 °C	Biochar from 775 °C	EBC Standard
MC	%	4.44 ± 2.87	4.08 ± 2.92	4.41 ± 2.33	4.00 ± 2.50	>60
OM		39.47 ± 2.30	33.00 ± 2.88	37.39 ± 3.05	24.60 ± 3.12	- *
pH	-	12.04 ± 0.02	13.24 ± 0.03	12.55 ± 0.14	13.40 ± 0.12	6–10
N	%	3.73 ± 0.02	3.05 ± 0.01	3.07 ± 0.02	3.69 ± 0.04	-
TOC		30.76 ± 0.02	29.89 ± 0.01	30.56 ± 0.03	30.29 ± 0.02	20
C/N	-	8.18	9.81	9.95	9.87	-
Ca	mg/kg	1469.00 ± 0.02	1466.64 ± 0.03	1403.38 ± 0.04	1487.14 ± 0.02	-
K		324.80 ± 0.04	359.00 ± 0.05	301.98 ± 0.05	266.72 ± 0.06	-
Mg		100.12 ± 0.03	112.44 ± 0.02	100.67 ± 0.03	93.65 ± 0.07	-
Na		281.70 ± 0.02	341.69 ± 0.01	271.07 ± 0.06	263.59 ± 0.02	-
P		1927.61 ± 0.02	1902.34 ± 0.09	1723.31 ± 0.08	1546.66 ± 0.04	-
S		95.86 ± 0.14	255.25 ± 0.12	238.74 ± 0.22	298.46 ± 0.02	-
Al		35.76 ± 0.02	- *	14.76 ± 0.04	25.12 ± 0.03	-
Cd		0.80 ± 0.03	0.3 ± 0.02	0.3 ± 0.03	0.4 ± 0.03	1.5
Co		0.33 ± 0.02	-	1.60 ± 0.02	0.36 ± 0.03	-
Cr		0.12 ± 0.02	-	-	-	100
Cu		0.55 ± 0.04	0.12 ± 0.03	-	0.46 ± 0.03	200
Fe		3.88 ± 0.03	3.19 ± 0.04	4.25 ± 0.04	2.13 ± 0.03	-
Mn		-	-	-	-	-
Ni		2.43 ± 0.02	0.80 ± 0.01	-	3.97 ± 0.02	50
Pb		-	3.65 ± 0.02	-	-	120
Zn		2.25 ± 0.03	2.36 ± 0.04	1.96 ± 0.02	1.53 ± 0.04	600

*—No data available.

**Table 6 materials-16-06314-t006:** Physicochemical characteristics of the investigated growing media prior to the plant growth experiment.

Growing Media	Units	A	B	C	D	E	F
Parameters		S	S + C1	S + C2	S + B	S + C1 + B	S + C2 + B
N	%	0.08 ± 0.02	0.14 ± 0.02	0.14 ± 0.02	0.11 ± 0.04	0.17 ± 0.03	0.18 ± 0.02
C		0.80 ± 0.02	1.63 ± 0.04	1.70 ± 0.06	1.04 ± 0.03	1.88 ± 0.04	2.01 ± 0.02
MC		1.54 ± 2.02	5.44 ± 0.12	4.46 ± 1.32	3.92 ± 2.92	3.69 ± 3.02	3.38 ± 2.09
OM		2.56 ± 3.02	4.57 ± 2.32	7.27 ± 3.08	3.91 ± 2.32	4.20 ± 2.09	4.19 ± 2.52
pH	-	6.99 ± 0.02	7.13 ± 0.02	7.16 ± 0.03	8.10 ± 0.02	7.62 ± 0.06	7.53 ± 0.04
C/N		10.00	11.64	12.14	9.45	11.06	11.17
P	mg/kg	131.90 ± 0.02	247.99 ± 0.02	236.47 ± 0.03	210.72 ± 0.01	292.30 ± 0.07	381.88 ± 0.04
S		51.78 ± 0.04	98.46 ± 0.02	95.16 ± 0.02	70.07 ± 0.03	121.08 ± 0.04	146.93 ± 0.04
Ca		826.56 ± 0.02	1127.26 ± 0.02	1341.55 ± 0.01	1092.16 ± 0.06	1275.51 ± 0.03	1769.16 ± 0.03
Mg		595.93 ± 0.02	598.18 ± 0.02	678.12 ± 0.08	641.36 ± 0.03	599.03 ± 0.07	698.02 ± 0.04
Na		3719.8 ± 0.02	2440.34 ± 0.04	2828.54 ± 0.04	3394.5 ± 0.08	2359.7 ± 0.07	2468.82 ± 0.09
K		2396.14 ± 0.04	2098.92 ± 0.03	2329.64 ± 0.02	2387.12 ± 0.05	1979.20 ± 0.02	2281.28 ± 0.05

**Table 7 materials-16-06314-t007:** The growth of cherry tomatoes in the 6-week pot experiment.

Growing Media	Week 1	Week 2	Week 6
A	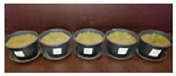	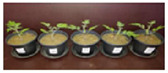	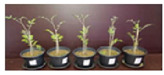
B	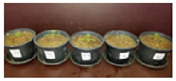	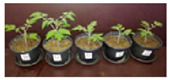	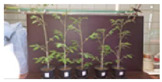
C	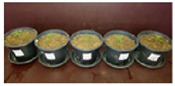	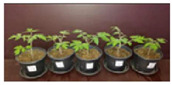	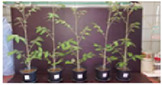
D	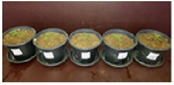	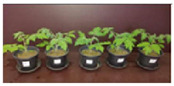	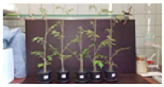
E	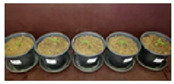	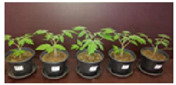	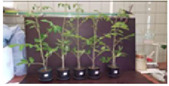
F	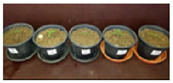	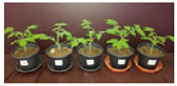	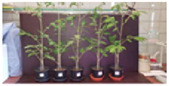

**Table 8 materials-16-06314-t008:** Physicochemical characteristics of the investigated growing media after the completion of the plant growing experiment.

Parameters	pH	N	C	MC	OM	C/N	Ca	K	Mg	Na	P
Units	-	%				-	mg/kg				
A	7.06 ± 0.02	0.06 ± 0.02	0.87 ± 0.02	2.26 ± 2.12	2.82 ± 3.15	14.55	293.8 ± 0.10	263.2 ± 0.21	183.3 ± 0.10	50.0 ± 0.12	46.5 ± 0.10
B	7.58 ± 0.02	0.13 ± 0.03	1.44 ± 0.06	2.51 ± 2.02	2.00 ± 3.19	11.32	524.1 ± 0.12	282.2 ± 0.22	189.3 ± 0.11	52.6 ± 0.18	72.1 ± 0.21
C	7.55 ± 0.06	0.12 ± 0.03	1.43 ± 0.07	2.62 ± 2.98	2.02 ± 2.45	12.03	405.7 ± 0.19	299.9 ± 0.21	184.4 ± 0.11	56.5 ± 0.15	74.1 ± 0.23
D	7.85 ± 0.08	0.08 ± 0.04	1.01 ± 0.13	1.42 ± 2.89	5.41 ± 2.19	12.27	637.6 ± 0.17	260.3 ± 0.25	184.9 ± 0.19	55.9 ± 0.14	69.0 ± 0.35
E	7.56 ± 0.08	0.11 ± 0.06	1.75 ± 0.13	1.52 ± 2.78	4.67 ± 2.28	15.38	565.0 ± 0.12	300.8 ± 0.28	198.1 ± 0.16	59.7 ± 0.30	86.5 ± 0.34
F	7.74 ± 0.09	0.14 ± 0.03	1.68 ± 0.12	1.02 ± 3.03	4.92 ± 2.12	12.29	619.1 ± 0.14	320.3 ± 0.30	186.1 ± 0.10	55.0 ± 0.24	89.7 ± 0.45

**Table 9 materials-16-06314-t009:** Chemical composition of the collected cherry tomatoes after the completion of the plant-growing experiment.

Parameters	N	C	C/N	Ca	K	Mg	Na	P
Units	%		-	mg/kg				
Plant—A	2.01 ± 0.02	36.94 ± 0.01	17.83	3413.6 ± 0.12	2696.6 ± 0.34	384.6 ± 0.35	397.4 ± 0.12	553.1 ± 0.43
Plant—B	2.13 ± 0.02	37.15 ± 0.02	17.46	2172.6 ± 0.22	3009.2 ± 0.43	364.2 ± 0.36	497.2 ± 0.12	519.4 ± 0.34
Plant—C	2.18 ± 0.02	36.02 ± 0.02	16.49	2491.6 ± 0.23	3369.6 ± 0.42	358.8 ± 0.22	530.6 ± 0.12	595.2 ± 0.45
Plant—D	2.62 ± 0.04	38.18 ± 0.02	23.51	2193.4 ± 0.32	2339.6 ± 0.25	276.8 ± 0.42	226.4 ± 0.46	350.8 ± 0.46
Plant—E	2.07 ± 0.03	36.86 ± 0.03	17.79	1879 ± 0.35	3683.8 ± 0.23	356.1 ± 0.25	621.1 ± 0.42	421.2 ± 0.32
Plant—F	2.30 ± 0.02	36.81 ± 0.01	16.03	1694 ± 0.23	3198.8 ± 0.32	327.2 ± 0.43	383.4 ± 0.35	379.4 ± 0.22

## Data Availability

Not applicable.

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
