# Peer review of "The Influence of Poultry Manure-Derived Biochar and Compost on Soil Properties and Plant Biomass Growth"

_materials, 2023, doi:10.3390/ma16186314_

Round 1

Reviewer 1 Report

It is my pleasure to revise your paper titled: '' The influence of poultry manure-derived biochar and composts on soil properties and plant biomass growth''. In my opinion, this work is original and interesting. 

The following are my comments and critique.

1- In the introduction, several sentences are referenced, please add the appropriate reference for each sentences.

2- In the introduction it is necessary to introduce the objective of your work.

Author Response

Dear Reviewer,

I would like to thank you for reviewing my manuscript. 

**Below, are my responses to your comments:

a) 1- In the introduction, several sentences are referenced, please add the appropriate reference for each sentences.

Thank you very much for your suggestion. I agree that this would be a good idea. I made changes to the text according to your suggestions.

b) 2- In the introduction, it is necessary to introduce the objective of your work.

 Thank you for pointing this out. Yes, I agree. I made changes in the introduction according to your suggestions. Also, in the introduction, I added the overall goal of this work which was to investigate the potential of poultry manure as a source to produce organic soil enhancers such as poultry manure-derived biochar and poultry manure-derived compost and to determine their physicochemical properties and effects on soil properties and growth of cherry tomatoes.

Best regards,

Danuta Dróżdż

Reviewer 2 Report

I reviewed the article on tittle ” The influence of poultry manure-derived biochar and composts 2 on soil properties and plant biomass growth” but its needs major revision before publication, see below some comment for improvements.

1.       Line 12-13:” manure which is “needs revised

2.       Line 15-16:” manure (PM) such as the “needs revised

3.       Line 21-22:” was characterized by the relatively “needs revised, like this the whole manuscript full of grammatical errors, so needs carefully revision

4.       Figure and should be self-explanatory

5.       Figure 9 should be revised

6.       Conclusions should be summarized

7.       Used updated references  

 Extensive editing of English language required

Author Response

Dear Reviewer,

I would like to, thank you for all your comments and recommendation. 

**Below, are my responses to your comments:

a) Line 12-13:” manure which is “needs revised.

Thank you for your suggestion, I added the corrections. 

b) Line 15-16:” manure (PM) such as the “needs revised.

Thank you for your comment. I added the corrections. 

c) Line 21-22:” was characterized by the relatively “needs revised, like this the whole manuscript full of grammatical errors, so needs careful revision.

Thank you for your comment. I added the corrections. Also, me and co-authors with consultation from the native speaker checked the grammatical correctness of the manuscript.

d) Figure and should be self-explanatory

Thank you, I added only one photo for a clearer overview of the layout of the cherry tomatoes in the phytotron chamber during the plant growth experiment.

e) Figure 9 should be revised.

Thank you for your suggestion. Based on the temperature evolution, the composting process of poultry manure with wheat straw proceeded in a proper way – typical for laboratory composting in closed vessels with forced aeration. Figure 9 was prepared with the results obtained during daily measurement of temperature inside the reactor and for comparison ambient temperature. I also changed the name of Figure 9 for a better understanding of the process.

f) Conclusions should be summarized.

Thank you for your suggestion. I removed unnecessary sections in the conclusions. The main conclusions that characterize fertilizer products such as nitrogen content, carbon content, phosphorus content, microbiological safety, and limit for heavy metals remained. These parameters are also included in the Fertilizing Product Regulation from July 2022 and are the key elements that can verify composts and biochar prepared from poultry manure as soil enhancers. 

g) Used updated references.

Thank you for your comment. I tried to use updated references for this manuscript because primary knowledge about poultry manure is well-known. However, the selection of biochar from the lowest pyrolysis temperature (475°C) was determined by the fact that there is limited knowledge about the effects of biochar from poultry manure obtained at the temperature 475°C on soil properties and plant growth. The knowledge about biochar is considered the youngest field in which the articles are published. The first articles began to appear at the turn of 2007-2009. This manuscript also contributes to increasing knowledge about the use of biochar from lower temperatures as soil enhancers. We report that biochar obtained at 475°C can be a beneficial fertilizer product, is microbiologically safe, and does not exceed the limits of heavy metals compared to the Fertilizing Product Regulation from 2022. The soil enhancers obtained, in the form of biochar and composts from poultry manure, fulfill the requirements according to the Fertilizing Product Regulation.

Best regards,

Danuta Dróżdż

Reviewer 3 Report

Manuscript ID: materials-2600654

Title: The influence of poultry manure-derived biochar and composts on soil properties and plant biomass growth

The authors investigated the potential of poultry manure, in form of poultry manure-derived biochar, respective poultry manure-derived compost, as source of organic soil enhancers. The effect of soil enhancers on soil properties and growth of cherry tomatoes was studied.

The manuscript is interesting and generally well-written. In my opinion, it can be published after a minor revision.

I have the following observations:

1.     In Abstract – “367 mln laying hens in 2021 and 7.2 mld broilers”. In “Introduction” – “360 million laying hens and 7.2 billion broilers”. The abbreviations must be avoided in Abstract, and a unitary mention is necessary.

2.     “Table 1. Selected characteristics of fresh poultry manure.” - It is not clear how the results were obtained. The authors must mention if they are results obtained by them or from the literature.

3.     L142 – “The dry bulk density of the straw was 120 kg/m3 and the air-filled porosity was 54%.” - The authors must mention if there are results obtained by them for the studied material and indicate the name of the method, especially since the methods are only described below. The same for “The wheat straw had a high C/N ratio of 82”. It would be better not to present experimental results before describing the methods used, to be understood more easily by readers.

4.     The “Conclusions” section can be shorter, with more concrete information about the results obtained in this study.

5.     Typos must be corrected: “the obtained the results”; L216 “The substrate were”; L368 “the hygienization of the composts were performed”, etc.

Author Response

Dear Reviewer, 

I would like to thank you for reviewing my manuscript. All the comments and recommendations are very valuable to me and will help to improve my corrections in manuscripts.  

**Below, are my responses to your comments and suggestions:

a) In Abstract – “367 mln laying hens in 2021 and 7.2 mld broilers”. In “Introduction” – “360 million laying hens and 7.2 billion broilers”. The abbreviations must be avoided in the Abstract, and a unitary mention is necessary.

Thank you very much, I agree with your comment and corrected the abstract for a better understanding of the overall goal of the manuscript. 

b) “Table 1. Selected characteristics of fresh poultry manure.” - It is not clear how the results were obtained. The authors must mention if they are results obtained by them or from the literature.

Thank you for your comment. Fresh poultry manure was collected by me from a caged poultry farm (from Cieszyn, Poland) and analyzed for physicochemical properties as I mentioned in the manuscript. I added more clear information in the text about this.

c)  L142 – “The dry bulk density of the straw was 120 kg/m3 and the air-filled porosity was 54%.” - The authors must mention if there are results obtained by them for the studied material and indicate the name of the method, especially since the methods are only described below. The same for “The wheat straw had a high C/N ratio of 82”. It would be better not to present experimental results before describing the methods used, to be understood more easily by readers.

Thank you for your suggestion. I added the missing knowledge about bulk density in the section Methods. I agree with you, but by presenting the basic physicochemical parameters at the beginning, I wanted to highlight the substrates that were used. So that readers could immediately get an overview of the substrates that were used in the experiment. In the results section I wanted to focus on the results already in the form of products, i.e. composts and biochar based on poultry manure. 

d) The “Conclusions” section can be shorter, with more concrete information about the results obtained in this study.

Thank you for your suggestion. I added the changes. I left the parts important for comparing the results obtained to the Fertilizing Product Regulation from July 2022 on soil enhancers. The soil enhancers obtained, in the form of biochar and composts from poultry manure, fulfill the requirements according to the Fertilizing Product Regulation. This is also an important topic for me, as I have been working with poultry manure for 5 years and have a great database of information about this substrate and their processing possibilities through thermal and biological processes. Also, in the future, I will try to focus more on specific data, as suggested in the comment.

e) Typos must be corrected: “the obtained the results”; L216 “The substrate were”; L368 “the hygienization of the composts were performed”, etc.

Thank you very much for the suggestion, I added the corrections.

Best regards,

Danuta Dróżdż

Round 2

Reviewer 2 Report

The author improved the manuscript according to my suggestions, so it's acceptable for publication.